Analysis of the genetic diversity of the coastal and island endangered plant species Elaeagnus macrophylla via conserved DNA-derived polymorphism marker

Wang Yi
Ma Yan
Jia Bingyu
Wu Qichao
Zang Dekui zangdk@sdau.edu.cn
Yu Xiaoyan
College of Forestry, Key Laboratory of State Forestry Administration for Silviculture of the Lower Yellow River, Shandong Agricultural University , Tai’an , Shandong province , China
Tatarinova Tatiana
Electronic publication date: 2020 Jan 31
Publication date: 2020
Volume: 8
Electronic Location ID: e8498
Received 2019 Jul 9; Accepted 2019 Dec 31
Copyright: ©2020 Wang et al.
Copyright year: 2020
Copyright holder: Wang et al.
License: This is an open access article distributed under the terms of the Creative Commons Attribution License, which permits unrestricted use, distribution, reproduction and adaptation in any medium and for any purpose provided that it is properly attributed. For attribution, the original author(s), title, publication source (PeerJ) and either DOI or URL of the article must be cited.
License URL: https://creativecommons.org/licenses/by/4.0/

Keywords: Elaeagnus macrophylla, CDDP markers, Conservation implications, Genetic variation

Funding: Shandong Agricultural Seeds Engineering Project 2019LZGC01802 Shandong Provincial Agricultural Elite Varieties Project 2016LZGC038 Forestry Science & Technology Innovation Project of Shandong Province LYCX01-2018-03 This study was funded by the Shandong Agricultural Seeds Engineering Project (Collection, Protection and Precision Identification of Forest Tree Germplasm Resources, 2019LZGC01802), the Shandong Provincial Agricultural Elite Varieties Project (2016LZGC038) and the Forestry Science & Technology Innovation Project of Shandong Province (LYCX01-2018-03), China. There was no additional external funding received for this study. The funders had no role in study design, data collection and analysis, decision to publish, or preparation of the manuscript.

==============================
The genetic diversity and genetic structure of five natural populations of the island and coastal endangered plant species Elaeagnus macrophylla were analyzed via conserved DNA-derived polymorphism molecular markers. A total of 289 discernible loci were obtained from 102 individuals via fifteen primers, and 100% of the loci were polymorphic. The observed number of alleles was 1.9654, and the effective number of alleles was 1.2604. Nei’s genetic diversity index was 0.1724 on average, and Shannon’s information index was 0.2869, indicating that Elaeagnus macrophylla had lower levels of genetic diversity than those reported for its continental relatives and other continental species. The average percentage of polymorphic loci was 42.1%, and the maximum and minimum were 80.97% and 14.88%, respectively, which were associated with the Nanji Island and Liugong Island populations, respectively. The populations of Elaeagnus macrophylla were highly differentiated. Cluster analysis revealed that the similarity between the tested samples was related to their geographical location, that the samples from the same island tended to cluster together, and that there was no cross-clustering between samples. The Nanji Island and Da Rushan populations differentiated into two subpopulations. Last, we detected no correlation between genetic distance and geographic distance between populations (Pearson’s correlation coefficient r = 0.256579, p-value = 0.8309).

Introduction

Elaeagnus macrophylla is an endangered evergreen shrub species of East Asian coastal areas and islands. It is distributed in the Shandong, Zhejiang, and Jiangsu Provinces of China, mainly on offshore islands and in coastal lowlands (Chinese Flora Editorial Board of the Chinese Academy of Sciences, 1983). Owing to its unique geographical distribution pattern, E. macrophylla is highly valuable for studying coastal flora and can be widely used in coastal greening because of its tolerance to sea breeze, salinity, drought, and poor soil (Zang, 2016). This species also has potential economic value; for example, it can be used for the production of fruit juice and wine (Zang, 2016). In recent years, with the rapid development of the economy and coastline, intensification of human interference, and continuous reduction in suitable environments, the number, and size of natural populations have decreased sharply, causing this species to become endangered.

The genetic diversity of island species is generally lower than that of continental species, the risk of extinction is greater for the former (Raven, 1998). From the 17th century to the 20th century, 384 species of vascular plants went extinct worldwide, 139 of which were island plant species. Moreover, forty percent of vulnerable or endangered vascular plant species are island species (Reid & Miller, 1989). Human disturbances, such as habitat destruction and invasion by alien species, are considered to be the main factors threatening island species (Wolf & Harrison, 2001). Studies of Ilex integra based on inter simple sequence repeat (ISSR) molecular markers (Leng et al., 2005) and Neolitsea sericea based on random amplified polymorphic DNA (RAPD) molecular markers (Wang et al., 2004) showed that the geographical isolation of islands had a significant effect on the genetic differentiation of island populations and that the genetic diversity of island relatives was lower than that of close continental relatives. However, no studies of the genetic diversity of the island plant species E. macrophylla have been conducted.

The conserved DNA-derived polymorphism (CDDP) method is based on a single primer amplification reaction, with primers designed to target conserved sequences of plant functional genes, mostly transcription factors such as WRKYs, MYBs, MADs, ERFs, KNOXs, and ABP1. Because of the strong conservation of some sequences of plant DNA, CDDP molecular marker technology can be used across different species. Studies of rice (Oryza sativa) have shown that CDDP markers have many advantages, including convenience, low cost, and rich polymorphism, and can effectively mark sequences of target traits (Collard & Mackill, 2009). Compared with traditional DNA molecular methods, the CDDP method is practical because the primers used in CDDP are specific for conserved gene sequences. By amplifying these conserved sequences, which tend to be linked to phenotypic traits, CDDP markers can provide advantages for plant genetic diversity assessment (Andersen & Lübberstedt, 2003). Since Poczai first successfully used CDDP markers and within-target markers to investigate the genetic diversity and group structure of Solanum dulcamara (Poczai et al., 2011), CDDP markers have proven useful in the analysis of several other plant species, such as Chrysanthemum (Li et al., 2014), Paeonia suffruticosa (Li, 2013), Vaccinium vitis-idaea (Fang et al., 2016), and Rosa rugosa (Jiang & Zang, 2018). However, CDDP markers have not yet been used to study E. macrophylla.

In this study, CDDP markers were used to analyze the genetic diversity and genetic relationships between major natural populations of E. macrophylla in China, with the aim of revealing the level of genetic diversity and degree of genetic differentiation, assessing the relationships between populations, examining the influence of geographical isolation and human factors on the genetic structure, and providing a scientific basis for the protection and rational utilization of E. macrophylla.

Materials and Methods

Plant materials

A total of 102 individual leaf samples were collected from 7 islands and offshore sites (Fig. 1 and Table 1) from April to July 2018; sampling was conducted within the natural distribution area of E. macrophylla. Interval sampling was applied except within small populations (such as the Liugong Island population, where samples from all individual plants found were collected). Only one individual each was found on Lingshan Island and Putuo Island. After the samples were collected, silica gel was used to quickly dry the specimens, after which they were stored at 20 °C.

Figure 1 Geographical location of the seven sampling points (including Putuo Island and Lingshan Island where only one sample was collected) of E. macrophylla in China.

Show the geographical location of seven sampling points by zooming in on the four areas A, B, C, and D.

Table 1 Sampling information for E. macrophylla.

The table shows information such as population name, abbreviation, geographic location, latitude and longitude, and altitude.

Population	Population abbreviation	Number	Locality	Geographical location	Altitude (meters)	
Liugong Island	LGD	8	Weihai Bay, Weihai city, Shandong Province	37°30′N, 122°10′E	22	
Da Rushan	DRS	20	Rushan city, Weihai city, Shandong Province	36°45′N, 121°30′E	5.2	
Lingshan Island	LSD	1	Huangdao District, Qingdao city, Shandong Province	36°27′N, 121°58′E	20	
Daguan Island	DGD	26	Laoshan District, Qingdao city, Shandong Province	36°13′N, 120°46′E	11	
Laoshan	LS	13	Laoshan District, Qingdao city, Shandong Province	36°7′N, 120°39′E	20	
Putuo Island	PTD	1	Zhoushan Islands, Zhoushan city, Zhejiang Province	30°0′N, 122°24′E	96	
Nanji Island	NJD	33	Pingyang County, Wenzhou city, Zhejiang Province	27°28′N, 121°3′E	42	

DNA extraction and PCR amplification

Total DNA was extracted from E. macrophylla via the modified cetyltrimethylammonium bromide (CTAB) method (Doyle & Doyle, 1987). The quality and purity of DNA were determined by 2% agarose gel electrophoresis, and a spectrophotometer (Thermo Fisher Scientific Inc., USA) was used to ensure DNA quantification. All DNA samples were stored at 20 °C for later use.

The DNA from one sample per population was selected to screen 21 CDDP primers (Collard & Mackill, 2009) (synthesized by Sangon Biotech, China). The results revealed 15 primers with clear and reproducible amplification bands were screened out (Table 2). PCR was conducted in a total reaction volume of 20 µl consisting of 10 µl of 2 × Ex Taq MasterMix (dye), 7.5 µl of double-distilled H2O, 1 µl of 30 ng/µl DNA template, and 1.0 µl of 10 pmol/µl primers (Sangon Biotech, China). A standard PCR thermocycler (RT-PCR 7500, Thermo Fisher Scientific Inc., USA) was used and the PCR program was as follows: an initial denaturation step at 94 °C for 3 min; 35 annealing cycles of 94  °C for 1 min, 50 °C for 1 min and 72 °C for 2 min; and a final extension of 72  °C for 5 min. The PCR products were subsequently stored at 4 °C. The products were then electrophoresed on a 2% agarose gel (110 V and 110 mA) for 1.5–2 h; a DL2000 marker was used as a size marker. The electrophoresis results were imaged and recorded by a gel imaging system. All amplification procedures were repeated at least twice to ensure the repeatability of the experiment.

Table 2 Site information for 15 CDDP markers and genetic diversity parameters at each locus of E. macrophylla.

Table contains inforations such as primer code, primer names, sequence, annealingTemperature, number of bands recorded, number of polymorphic bands and the percentage of polymorphism.

Primer code	Primer name	Sequence (5′–3′)	Annealing temperature	Number of bands recorded	Number of polymorphic bands	The percentage of polymorphism/%	
Pr1	WRKY-F1	TGGCGSAAGTACGGCCAG	50	21	21	100	
Pr2	WRKY-R1	GTGGTTGTGCTTGCC	52	30	30	100	
Pr3	WRKY-R3	CCGCTCGTGTGSACG	50	21	25	100	
Pr4	MYB1	GGCAAGGGCTGCCGC	50	19	19	100	
Pr5	MYB2	GGCAAGGGCTGCCGG	50	13	13	100	
Pr6	ERF1	CACTACCGCGGSCTSCG	50	30	30	100	
Pr7	ERF2	GCSGAGATCCGSGACCC	50	11	11	100	
Pr8	ERF3	TGGCTSGGCACSTTCGA	50	11	11	100	
Pr9	KNOX-1	AAGGGSAAGCTSCCSAAG	50	21	21	100	
Pr10	KNOX-2	CACTGGTGGGAGCTSCAC	50	19	19	100	
Pr11	KNOX-3	AAGCGSCACTGGAAGCC	50	15	15	100	
Pr12	MADS-1	ATGGGCCGSGGCAAGGTGC	50	14	14	100	
Pr13	MADS-4	CTSTGCGACCGSGAGGTG	50	28	28	100	
Pr14	ABP1-1	ACSCCSATCCACCGC	50	14	14	100	
Pr15	ABP1-3	CACGAGGACCTSCAGG	50	18	18	100	

Statistics and data analysis

We used POPGEN v.1.32 (Yeh, Yang & Boyle, 1999) to compute the following parameters: observed allele number (Na), effective allele number (Ne), Nei’s genetic diversity index (H), Shannon’s information index (I), polymorphic loci, percentage of polymorphic loci (PPL, %), total genetic diversity (Ht), genetic diversity within populations (Hs), the genetic differentiation coefficient (Gst) and gene flow (Nm) between populations. Estimates were also calculated within sampling localities when significant differences among specimens were detected.

A dendrogram was generated by the unweighted pair group method with arithmetic mean (UPGMA) clustering procedure in NTSYS-pc 2.10e software (Rohl, 1994). The relationship between geographic distance and genetic distance was analyzed with Pearson’s correlation coefficients in R.

The genetic structure of populations was further assessed via the Bayesian clustering approach implemented in STRUCTURE v.2.3.4 (Pritchard, Stephens & Donnelly, 2000). The number of potential genetic clusters (K values) was set from 1 to 10, with 10 independent runs for each K. The contribution of the accessions to the genotypes was calculated on the basis of a 105 iteration burn-in period and 105 iteration sampling period. The optimal number of K clusters was then identified according to the methods of Evanno, Regnaut & Goudet (2005).

Results and Analysis

Population- and species-level diversity of E. macrophylla

The DNA of 102 samples was amplified with 15 primers, yielding 289 bands, and the fragment length was between 500 and 2,000 bp (Fig. 2). The number of amplified bands ranged from 11 to 30, and the average number of amplified bands was 19.3. The number of Pr2 and Pr6 amplification bands was the highest, at 30, and the number of Pr7 and Pr8 amplification bands was the lowest, at 11. The percentage of polymorphisms reached 100% (Table 2), which indicated that the genomic DNA polymorphism of E. macrophylla was high.

Figure 2 Amplification results for MYB1 in the Liugong Island population and the Da Rushan population.

Amplification results for MYB1 in the Liugong Island population (1–8) and the Da Rushan population (9–20), Marker=DL2000.

At the population level, the PPL ranged from 14.88% to 80.28%, with an average of 48.928%, whereas it was 96.54% at the species level. The Na ranged from 1.1488 to 1.8028, while the Ne varied from 1.0739 to 1.2410. H varied from 0.0446 to 0.1580, with an average of 0.1149, and I ranged from 0.0690 to 0.2613, with an average of 0.1848. At the species level, H and I were 0.1724 and 0.2869, respectively (Table 3). The Na, Ne, H, I, and PPL were consistent among populations, with the NJD population presenting the largest values and the LGD population presenting the lowest values, all of which were lower than those at the species level.

Table 3 Genetic diversity in five populations of E. macrophylla.

The table contains information such as population name, number of samples, Na, Ne,H, I, PPL, and standard deviation in parentheses.

Population name	Number of samples	Na	Ne	H	I	PPL (%)	
LGD	8	1.1488 (0.3565)	1.0739 (0.2178)	0.0446 (0.1210)	0.0690 (0.1785)	14.88	
DRS	20	1.5398 (0.4993)	1.2002 (0.2850)	0.1290 (0.1622)	0.2070 (0.2381)	53.98	
NJD	33	1.8028 (0.3986)	1.2410 (0.2931)	0.1580 (0.1566)	0.2613 (0.2198)	80.28	
LS	13	1.4998 (0.4983)	1.1941 (0.3020)	0.1208 (0.1680)	0.1898 (0.2458)	50.52	
DGD	26	1.5502 (0.4983)	1.1912 (0.2878)	0.1222 (0.1615)	0.1968 (0.2355)	44.98	
Mean	20	1.5083	1.1801	0.1149	0.1848	48.928	
Species level	100	1.9654 (0.1831)	1.2601 (0.2845)	0.1724 (0.1532)	0.2869 (0.2098)	96.54	
Notes.

The PTD and LSD populations were not included because there was only one sample for each.

Genetic differentiation of the populations of E. macrophylla

The Ht and Hs were 0.1706 and 0.1149, respectively, as calculated by POPGEN v.1.32 software. The Gst was 0.3263, indicating that 67.37% of the variation was within the populations and that 32.63% of the variation occurred between the populations. A certain degree of genetic differentiation was observed between the populations. The Nm was 1.0325, indicating that there was some (albeit limited) genetic exchange between populations.

Genetic differentiation between populations can be further analyzed on the basis of Nei’s genetic distance and genetic identity. For the five populations of E. macrophylla, the genetic distance was between 0.0490 and 0.1443 (Table 4), with a mean of 0.08127, and Nei’s genetic identity was between 0.8656 and 0.9588, with a mean of 0.9226.

Table 4 Nei’s genetic identity (above diagonal) and genetic distance (below diagonal) for five populations.

Liugong Island (LGD), Da Rushan (DRS), Nanji Island (NJD), Laoshan (LS), Daguan Island (DGD).

Population	LGD	DRS	NJD	LS	DGD	
LGD	****	0.9253	0.8656	0.8730	0.8697	
DRS	0.0776	****	0.9588	0.9569	0.9431	
NJD	0.1443	0.0421	****	0.9522	0.9391	
LS	0.1358	0.0440	0.0490	****	0.9427	
DGD	0.1396	0.0585	0.0628	0.0590	****	

UPGMA cluster analysis

The applied measure of genetic similarity was used to construct UPGMA dendrograms (Fig. 3). The clustering map showed that the five populations could be divided into three groups. One group represented the LGD population. The DRS, NJD, and LS populations formed a second group, indicating that these three populations are closely related, the third group represented the DGD population. Populations with similar geographical distances were not clustered into the same group, indicating that the genetic distance between the populations of E. macrophylla was not related to geographical distance. Consistent with these results, the Pearson correlation coefficient test revealed no significant correlation between geographic and genetic distance (Pearson’s correlation coefficient r = 0.256579, p-value = 0.8309).

Figure 3 UPGMA cluster analysis of genetic similarity of 5 populations.

UPGMA cluster analysis of genetic similarity of five populations: LGD, DRS, NJD, LS, DGD.

Figure 4 UPGMA cluster analysis of 102 samples of E. macrophylla.

LGD (1-8), DRS (9-28), NJD (29-61), PTD (62), LSD (63), LS (64-76), DGD (77-102).

The UPGMA clustering map provided a clear division of the 102 samples (Fig. 4). Notably, cross-clustering occurred between samples from different populations, and samples from the same island tended to cluster together. The groups in the clustering results were generally consistent with the regional sources of the samples. As clearly shown by the clustering map, all the samples from the LGD population formed a small branch and then formed another branch with all the samples from the DRS population. Among the samples, Nos. 9–18 from the DRS population were more closely related to the samples from the LGD population than to Nos. 19–28 from the DRS population; thus, differentiation within the DRS population was observed between Nos. 9–18 and Nos. 19–28. All samples from the LS population formed a group, all samples from the DGD population formed another group, and the two groups formed a large branch. One sample from PTD formed a separate group. Samples from the NJD population were composed of two subpopulations. On the basis of the sampling location and latitude and longitude, samples 29–40 were collected in the northwestern part of NJD (121°3′24–121°3′8, 27°27′53–27°28′21), and samples 41–61 were collected in the southeastern part of NJD (121°5′52–121°6′11, 27° 26′54–27°27′12). Twenty-one samples (Nos. 41–61) formed a group, 12 samples (Nos. 29–40) formed another group, and these two groups formed different branches. Therefore, the samples could be easily divided into five groups on the basis of the clustering results.

Population structure analysis

The results of the Bayesian clustering analysis of the genetic structure showed that the populations of E. macrophylla best fit three genetic groups, and when K = 5, the delta K value was also large (Fig. 5A). When K = 3 (Fig. 5B), the LGD population and half of the DRS samples clustered into the first groups, further indicating that the two populations were closely related. Most samples from the LS population, half of the samples from the DRS population and all the samples from the NJD population clustered into the second group, and a portion of the samples from the LS and DGD populations formed the third group. When K = 5 (Fig. 5C), the LGD population and half of the DRS samples clustered into the first group, half of the DRS samples and the NJD samples indicated in yellow in the figure clustered into the second group, the NJD samples indicated in blue in the figure formed the third group, all samples from the LS population clustered into the fourth group, and all samples from the DGD population clustered into the fifth group. The LGD, LS, and DGD population pedigrees were simple, and the samples from the islands tended to cluster together. However, the NJD and DRS populations were different. The NJD population formed two subpopulations when K = 5: a northwestern group (indicated in yellow) and a southeastern group (indicated in blue). The DRS population differentiated into two subpopulations when K = 3 and K = 5: Nos. 9–18 and Nos. 19–28, respectively.

Figure 5 Population structure analysis and Delta-K values.

(A) Delta-K values; (B) population structure analysis, K = 3; (C) population structure analysis, K = 5.

Gst and Nm of the two NJD and DRS subpopulations

The Nm between the NJD and DRS subpopulations was 2.6084 and 2.0843, respectively, the Gst was 0.1609 and 0.1935, respectively. The subpopulations presented strong gene flow but high genetic differentiation.

Discussion

Frankham (1997) compared and analyzed the allelic diversity of 202 groups of land and island populations of various species, including those of mammals, birds, fish, reptiles, insects, and plants: in 165 cases (81.7%), the genetic diversity of island populations was lower than that of continental populations, with an average decrease of 29%. The average PPL via the CDDP markers in five populations of the island plant species E. macrophylla was 48.928%, the Ne was 1.1801, H was 0.1149, and I was 0.1848. All these numbers are far below those of populations of the continental species Camellia japonica (PPL = 86.11%, Ne = 1.4775, H = 0.2940, I = 0.4459) (Juan, 2018) and Paeonia suffruticosa (PPL = 72.1%, Ne = 1.2389, H = 0.1623, I = 0.2682) (Li, 2013) estimated on the basis of CDDP marker data. Moreover, compared with the rich genetic diversity of another species within the same genus (Elaeagnus mollis) (Qin, Zhang & Yan, 2006), the genetic diversity of E. macrophylla is low, which is consistent with the results of Frankham. As an endangered coastal plant species, E. macrophylla has a lower genetic diversity than its continental relatives and other continental species have. For island species, breeding characteristics, dispersal capability, and effective population size are often considered important factors affecting genetic diversity (Frankham, 1997; Weller, Sakai & Straub, 1996). E. macrophylla is a typical bisexual flowering plant species that produces flowers that have a small diameter and that produce nectar; thus, this species relies mainly on small insects as pollinators for cross-pollination (Zang, 2012). The Ht of E. macrophylla was lower than the average value of insect-pollinated plants (Ht = 0.2019; Hanwick & Godt, 1990), indicating that its insect-borne pollination has been affected to some extent. Owing to the large sea breeze on the island, insects can live only in groups, which affects their range of activities. As a result, pollen transmission is limited to a very small range, and random fixation of alleles and limited gene exchange leads to poor population expansion, thus reducing genetic diversity (Hamrick & Nason, 1996; Hamilton & Miller, 2002). As another important carrier of gene flow, seeds are also essential for the natural regeneration and expansion of plant populations and for increasing their genetic diversity (Hamilton & Miller, 2002). According to previous studies, E. macrophylla can produce fruit naturally in the wild, and the fruit is sweet (Zang, 2016). The fruit is heavily favored by birds, and some birds that feed on these fruits can spread the seeds. Unfortunately, owing to the influence of insect pollination, the seed setting rate in the wild is low. Moreover, owing to increased amounts of human activities, especially the vigorous development of tourism, island birds are becoming increasingly rare, which further restricts the spread of seeds and affects the genetic diversity of populations. Compared with that of C. japonica and E. mollis, the distribution of populations of E. macrophylla, an island species, is small. E. macrophylla occupies a fragile habitat and has a narrow distribution range, making it more vulnerable than continental species to extinction (Francisco-Ortega et al., 2000). Island segregation means fragmentation of habitats, which leads to the maintenance of small populations in fragmented habitats; as such, genetic diversity of endangered species may be lost because genetic drift causes allele loss and because inbreeding frequencies are increasing (Zhang & Jiang, 1999; Emerson, 2002).

In the present study, the Ht and Hs of E. macrophylla were 0.1706 and 0.1149, respectively. Compared with endangered and Chinese secondary protected plants (Fu & Jin, 1992), including R. rugosa, in previous CDDP-based (Ht = 0.2770, Hs = 0.1522) studies (Jiang & Zang, 2018), E. macrophylla in the present study showed low diversity. Similarly, the diversity values in the present study were lower than those estimated via CDDP markers for the plant species C. japonica (Ht = 0.2874, Hs = 0.2518; Juan, 2018). Gst is calculated as the ratio of between-population genetic variance to the total variance among populations (Wright, 1965). The Gst of the five populations of E. macrophylla was 0.3263, indicating that 32.63% of the variation existed among the populations. Genetic differentiation of the E. macrophylla populations was significant on the basis of Nei’s Gst classification criteria for genetic differentiation (low, Gst <  0.05; medium, Gst = 0.05∼0.15; and high, Gst > 0.15) (Nei, 1978). This genetic differentiation value was greater than the average value of 23 species (28.06%) of the Carina Islands (Francisco-Ortega et al., 2000). Furthermore, the genetic differentiation between E. macrophylla populations was relatively high, and the prevention of gene flow, genetic drift, and inbreeding was the main cause of genetic differentiation among populations (Starkin, 1987; Ouborg, Piquot & Groenendael, 1999; Manel et al., 2003).

Nm refers to the process by which a biological individual disperses from its place of origin, followed by the exchange of genes between populations. Such exchange may occur between biological populations of the same species or between different species and is essential to the evolution of many plant populations (Grant, 1991; Gerber et al., 2014). The populations of E. macrophylla displayed little gene flow (Nm = 1.0325), and the UPGMA clustering analysis of samples revealed no hybridization among individuals from different localities. Structure analysis (K = 5) revealed that most of the populations had a simple pedigree, and the genetic exchange between each pair of populations was low. These results were mainly due to the geographical isolation of the islands (mainly barriers posed by seawater), which limited the range of dispersal by pollen- and seed-dispersing birds (Kwon & Morden, 2002). For the E. macrophylla populations, the shortest distance is between the populations of LS and DGD (8,100 m); it is difficult for small pollinators to spread pollen across islands separated by vast seas. In addition to pollen, seeds play an important role in the spread of gene flow. One important aspect of seed movement is its role in the initial founding of a population (Chung et al., 2002). In a study of N. sericea, owing to seeds being dispersed over long distances by birds, numerous seedlings and juveniles (mostly aged <5 yr) of N. sericea were dispersed within a range of approximately 480 m (Hakdongri on Keojae Island) and 680 m (Naechori on Oenaro Island), respectively, from their maternal plants (Chung et al., 2002). The fruit of E. macrophylla is a drupe-shaped nut with a seed and rich flesh (Zang, 2012). After birds feed and digest their food, they can excrete seeds in their feces. This type of transmission is called intra-animal transmission, which carries seeds far away and promotes gene exchange (Petit et al., 2003). It is difficult to directly observe how far birds can spread seeds in the wild, but the retention time of seeds in the digestive tract of fruit-eating birds can be used to determine the potential propagation distance and the ability to reach the appropriate breeding ground (Manson & Stiles, 1998). The retention time of fruits of Sorbus pohuashanensis in the digestive tract of birds were found to be approximately 20 min. The first stopping point after feeding was mostly located between 5 and 10 m from the female plants, but the birds could have had many various landing points within 20 min, which might spread the seeds to distant areas (Zhang et al., 2010). However, birds are affected by factors such as seed size, feed intake, digestion, and excretion; thus, it is very difficult to distribute seeds between distances over sea areas greater than 8000 m or farther. Last, water currents are also a major medium of genetic exchange between islands (Kwon & Morden, 2002). Zhang et al. (2007) studied the differences in genetic variation between the species E. emarginata, I. integra and Machilus thunbergii. Clustering analysis revealed that the reason for the intermixing of individuals among E. emarginata populations was that seeds floating with ocean currents promoted gene exchange among populations, while those of I. integra and M. thunbergii did not. Therefore, the genetic differentiation of E. emarginata is lower than that of I. integra and M. thunbergii. The fruits of E. macrophylla fall into the sea because of sea breezes, but the spread of seeds by ocean currents to allow genetic exchange between populations is a rare event, as submersion quickly reduces the germination capability after a few days (Angélique & Debussche, 2000). When soaked in seawater, seeds of I. integra also presented a germination rate close to zero (Leng et al., 2005). Our sample clustering map revealed no hybridization among individuals, indicating that the genetic exchange between the populations is limited. Therefore, it is very difficult for E. macrophylla to achieve genetic exchange via seeds floating by currents. For E. emarginata, the reason for success may be that the distance between the islands is short and the seeds do not lose their ability to germinate after floating.

The results of the Bayesian clustering analysis showed, that when K = 3, the same gene pool exists between different populations. On the basis of limited gene flow and high genetic differentiation between populations, we speculate that the reason for this may be that E. macrophylla inherits the gene pool of its ancestors, such that populations with distant geographic distances also are part of the same gene pool.

As opposed to the high genetic differentiation between different island populations, individuals on islands are often grouped into a single group, indicating that individuals in each population have relatively close kinship, which may be related to the small island range, similar habitats within the population, and strong gene flow (Sahuquillo & Lumaret, 1995; Carlos, Emerson & Oromi, 2000); however, the DRS and NJD populations are exceptions. According to the UPGMA clustering and STRUCTURE analysis results, the DRS population differentiated into two subpopulations: Nos. 9–18 and Nos. 19–28. Gene flow (Nm = 2.0843) was strong between the two subpopulations, but there was high genetic differentiation (Gst = 0.1935). Because gene flow was not blocked and the habitats were similar, we speculate that the cause of this result was that the DRS population contains two gene pools, and complete gene introgression has not yet occurred in the two subpopulations under the condition of a gene flow of 2.0843. When K = 3, the NJD population did not exhibit differentiation; when K = 5, the NJD population differentiated into two subpopulations: a northwestern group and a southeastern group. The gene flow between the two subpopulations was determined to be 2.6084, indicating strong genetic exchange between the two subpopulations of NJD. However, there was still some genetic differentiation between the two subpopulations (Gst = 0.1609). After possible causes such as the prevention of gene flow were excluded, the reason may be related to differences in habitats within the population. The NJD nature reserve has numerous islands and reefs with meandering shorelines, headlands, and numerous bays. There are many types of coastal beaches, such as mudflats, gravel beaches, and rocky reefs, and NJD is at the intersection of the Taiwan Warm Current and the Jiangsu and Zhejiang Coastal Currents. The flow system is complex, so the habitat is complex (Xiao, 2007). The species of marine shellfish algae in this area not only are abundant but also have characteristics beneficial to temperate zones and tropical zones. Moreover, an obvious regional “fracture distribution” phenomenon is occurring. It is rare that three species with characteristics beneficial to different climate zones (i.e., tropical, subtropical and temperate zones) coexist in the NJD sea area at the same time; for example, typical tropical species such as Oliva emicator can survive in the NJD above 27°N, which is sufficient to demonstrate the particularity of the NJD habitat (Xiao, 2007). The two subpopulations are separately located in the northwestern and southeastern regions of NJD, and may exhibit some genetic differentiation because of habitat differences. However, our current evidence is insufficient, and additional research is needed.

In the comparison of the genetic diversity index of 22 endemic plant species on the Canary Islands, the average genetic diversity index of island species with relatively large populations (number of individuals >2,500; H = 0.1460) was significantly greater than that of island species with relatively small populations (number of individuals <100; H = 0.0970) (Francisco-Ortega et al., 2000). The average diversity index of each population of E. macrophylla was (from large to small) NJD (0.1580)>DRS (0.1290)>DGD (0.1222)>LS (0.1208)>LGD (0.0446), with an average value of 0.1149. The average genetic diversity index of the NJD population is much greater than that of the LGD population. The values of the other three populations are relatively similar. The NJD population is the largest, the DRS, DGD, and LS populations are similar in size, and the LGD population is declining. Human disturbances such as excessive logging, habitat destruction, and the introduction of exotic species are considered to be the main causes endangering island species (Atkinson, 1989; Frankham, 1997; Raven, 1998), which are manifested mainly as effective population decline, increased frequencies of inbreeding, loss of genetic diversity, decline in survival competitiveness, etc. (Ferson & Burgman, 1995; Frankham, 1997; Mengens, 1998). NJD is far from mainland China, is relatively closed, and experiences relatively limited exchange with the mainland, so there is little anthropogenic damage. Moreover, the island area is large, and the genetic background is complex, so the island presents high genetic diversity and a large population. LGD is a famous tourist destination in China, and the coastline is developing rapidly. Moreover, LGD has a small area, and its populations have similar genetic backgrounds and a single habitat; as such, loss of genetic diversity and populations declines have occurred.

Conservation of E. macrophylla diversity

Am in situ conservation method was proposed because the conservation of sufficient natural population numbers and sizes to prevent a reduction in genetic diversity is urgently needed. The best strategy for in situ conservation of genetic diversity during an endemic is the preservation of natural habitat (Francisco-Ortega et al., 2000). In this study, the NJD population displayed relatively high genetic diversity and should, therefore, be a priority for in situ conservation. The LGD population had the lowest genetic diversity and the smallest population size; the site of this population should be protected as the most urgent site. Furthermore, natural protection areas should be established to conserve and restore the habitat and populations, the awareness of local residents and tourism management personnel should be heightened, and the populations should be increased by artificial cultivation and subsequent management; for example, seeds collected from other populations could be sown, branches could be collected for cuttings, gene barriers could be broken by appropriate species regression, and the genetic diversity of populations could be increased. Moreover, to achieve effective conservation of germplasm resources, efforts are needed to carefully plan and construct pollen banks and gene banks for E. macrophylla.

Conclusions

The present study is the first genetic investigation of Elaeagnus macrophylla using conserved DNA-derived polymorphism markers to investigate the distribution and genetic variation. The results showed that conserved DNA-derived polymorphism markers can be effectively used to study the genetic diversity of Elaeagnus macrophylla populations and revealed that Elaeagnus macrophylla populations have low genetic diversity and high genetic differentiation. The low levels of gene flow between populations are the main cause of the high levels of genetic differentiation. On the basis of these findings, some conservation measures for Elaeagnus macrophylla are proposed.

Supplemental Information

Supplemental Information 1 A 0-1 matrix of 15 primer-pair amplified 102 samples

0-1 matrix obtained from the results of electropherogram.

Click here for additional data file.

Supplemental Information 2 Screening of primers ERF2, ERF3, KNOX-1 by 7 samples

It can be seen that the three primer amplification bands are clear and specific and can be used for experiments.

Click here for additional data file.

Supplemental Information 3 Screening of primers MYB1, MYB2, ERF1 by 7 samples

It can be seen that the MYB1, MYB2, ERF1 amplified bands are clear and specific and can be used for experiments.

Click here for additional data file.

Supplemental Information 4 Screening of primers WRKY-F1, WRKY-R1, WRKY-R2 by 7 samples

It can be seen that the WRKY-F1, WRKY-R1 amplified bands are clear and specific and can be used for experiments. Although the WRKY-R2 has a strip, the strip is weak and therefore not used.

Click here for additional data file.

Supplemental Information 5 Screening of primers WRKY-R3, WRKY-R2B, WRKY-R3B by 7 samples

It can be seen that the WRKY-R3 amplified band is clear and specific, and can be used for experiments. WRKY-R2B, WRKY-R3B is not effective, so it is not used.

Click here for additional data file.

Supplemental Information 6 Screening of primers KNOX-2, KNOX-3, MADS-1 by 7 samples

It can be seen that KNOX-2, KNOX-3, MADS-1 amplified bands are clear and specific and can be used for experiments.

Click here for additional data file.

Supplemental Information 7 Screening of primers ABP1-1, ABP1-2, ABP1-3 by 7 samples

It can be seen that the ABP1-1, ABP1-3 amplified bands are clear and specific and can be used for experiments. ABP1-2 did not amplify the band and therefore could not be used.

Click here for additional data file.

Supplemental Information 8 Screening of primers MADS-2, MADS-3, MADS-4 by 7 samples

It can be seen that the MADS-4 amplification band is clear and specific and can be used for experiments. Although the MADS-2 has a band, it is not effective in use, and the MADS-3 does not amplify the band, so it is not used.

Click here for additional data file.

Supplemental Information 9 LGD1-8,DRS9-20,WRKY-F1

Amplification results of WRKY-F1 on LGD1-8 and DRS9-20.

Click here for additional data file.

Supplemental Information 10 DRS21-28,NJD1-12

Amplification results of WRKY-F1 on DRS21-28, NJD1-12.

Click here for additional data file.

Supplemental Information 11 NJD13-NJD32,WRKY-F1

Amplification results of WRKY-F1 on NJD13-NJD32.

Click here for additional data file.

Supplemental Information 12 NJD 33,LS 1-13,PTD1,LSD1,WRKF-F1

Amplification results of WRKY-F1 on NJD 33, LS 1-13, PTD1, LSD1.

Click here for additional data file.

Supplemental Information 13 DGD1-13,WRKY-F1

Amplification results of WRKY-F1 on DGD1-13.

Click here for additional data file.

Supplemental Information 14 DGD14-26,WRKY-F1

Amplification results of WRKY-F1 on DGD14-26.

Click here for additional data file.

Supplemental Information 15 LGD1-8,DRS9-20,WRKY-R1

Amplification results of WRKY-R1 on LGD1-8 and DRS9-20.

Click here for additional data file.

Supplemental Information 16 DRS21-28,NJD1-12,WRKY-R1

Amplification results of WRKY-R1 on DRS21-28, NJD1-12.

Click here for additional data file.

Supplemental Information 17 NJD13-32,WRKY-R1

Amplification results of WRKY-R1 on NJD13-32.

Click here for additional data file.

Supplemental Information 18 NJD 33,LS1-13,PTD1,LSD1,WRKY-R1

Amplification results of WRKY-R1 on NJD 33, LS1-13, PTD1, LSD1.

Click here for additional data file.

Supplemental Information 19 DGD1-20,WRKY-R1

Amplification results of WRKY-R1 on DGD1-20.

Click here for additional data file.

Supplemental Information 20 DGD21-26,WRKY-R1

Amplification results of WRKY-R1 on DGD21-26.

Click here for additional data file.

Supplemental Information 21 LGD1-8,DRS9-20,WRKY-R3

Amplification results of WRKY-R3 on LGD1-8, DRS9-20 samples.

Click here for additional data file.

Supplemental Information 22 DRS21-28,NJD1-12,WRKY-R3

Amplification results of WRKY-R3 on DRS21-28, NJD1-12 samples.

Click here for additional data file.

Supplemental Information 23 NJD13-32,WRKY-R3

Amplification results of WRKY-R3 on DRS21-28, NJD13-32 samples.

Click here for additional data file.

Supplemental Information 24 NJD33,LS1-13,PTD1,LSD1,WRKY-R3

Amplification results of WRKY-R3 on NJD33, LS1-13, PTD1, LSD1 samples.

Click here for additional data file.

Supplemental Information 25 DGD1-20,WRKY-R3

Amplification results of WRKY-R3 on DGD1-20 samples.

Click here for additional data file.

Supplemental Information 26 DGD21-26,WRKY-R3

Amplification results of WRKY-R3 on DGD21-26 samples.

Click here for additional data file.

Supplemental Information 27 LGD1-8,DRS9-20,MYB1

Amplification results of MYB1 on LGD1-8, DRS9-20 samples.

Click here for additional data file.

Supplemental Information 28 DRS21-28,NJD1-12,MYB1

Amplification results of MYB1 on DRS21-28, NJD1-12 samples.

Click here for additional data file.

Supplemental Information 29 NJD13-33,MYB1

Amplification results of MYB1 on NJD13-33 samples.

Click here for additional data file.

Supplemental Information 30 NJD33,LS1-13,PTD1,LSD1,MYB1

Amplification results of MYB1 on NJD33, LS1-13, PTD1, LSD1 samples.

Click here for additional data file.

Supplemental Information 31 DGD1-20,MYB1

Amplification results of MYB1 on DGD1-20 samples.

Click here for additional data file.

Supplemental Information 32 DGD21-26,MYB1

Amplification results of MYB1 on DGD21-26 samples.

Click here for additional data file.

Supplemental Information 33 LGD1-8,DRS9-20,MYB2

Amplification results of MYB2 on LGD1-8, DRS9-20 samples.

Click here for additional data file.

Supplemental Information 34 DRS21-28,NJD1-12,MYB2

Amplification results of MYB2 on DRS21-28, NJD1-12 samples.

Click here for additional data file.

Supplemental Information 35 NJD13-32,MYB2

Amplification results of MYB2 on NJD13-32 samples.

Click here for additional data file.

Supplemental Information 36 NJD33,LS1-13,PUD1,LSD1,MYB2

Amplification results of MYB2 on NJD33, LS1-13, PUD1, LSD1 samples.

Click here for additional data file.

Supplemental Information 37 DGD1-20,MYB2

Amplification results of MYB2 on DGD1-20 samples.

Click here for additional data file.

Supplemental Information 38 DGD21-26,MYB2

Amplification results of MYB2 on DGD21-26 samples.

Click here for additional data file.

Supplemental Information 39 LGD1-8,DRS9-20,ERF1

Amplification results of ERF1 on LGD1-8, DRS9-20 samples.

Click here for additional data file.

Supplemental Information 40 DRS21-28,NJD1-12,ERF1

Amplification results of ERF1 on DRS21-28, NJD1-12 samples.

Click here for additional data file.

Supplemental Information 41 NJD13-32,ERF1

Amplification results of ERF1 on NJD13-32 samples.

Click here for additional data file.

Supplemental Information 42 NJD33,LS1-13,PTD1,LSD1,ERF1

Amplification results of ERF1 on NJD33, LS1-13, PTD1, LSD1 samples.

Click here for additional data file.

Supplemental Information 43 DGD1-20,ERF1

Amplification results of ERF1 on DGD1-20 samples.

Click here for additional data file.

Supplemental Information 44 LS1-6,DGD21-26,ERF1

Amplification results of ERF1 on LS1-6, DGD21-26 samples.

Click here for additional data file.

Supplemental Information 45 LGD1-8,DRS9-20,ERF2

Amplification results of ERF2 on LGD1-8, DRS9-20 samples.

Click here for additional data file.

Supplemental Information 46 DRS21-28,NJD1-12,ERF2

Amplification results of ERF2 on DRS21-28, NJD1-12 samples.

Click here for additional data file.

Supplemental Information 47 NJD13-32,ERF2

Amplification results of ERF2 on NJD13-32 samples.

Click here for additional data file.

Supplemental Information 48 NJD33,LS1-13,PTD1,LSD1,ERF2

Amplification results of ERF2 on NJD33, LS1-13, PTD1, and LSD1 samples.

Click here for additional data file.

Supplemental Information 49 DGD1-20,ERF2

Amplification results of ERF2 on DGD1-20 samples.

Click here for additional data file.

Supplemental Information 50 DGD21-26,ERF2

Amplification results of ERF2 on DGD21-26 samples.

Click here for additional data file.

Supplemental Information 51 LGD1-8,DRS9-20,ERF3

Amplification results of ERF3 on LGD1-8, DRS9-20 samples.

Click here for additional data file.

Supplemental Information 52 DRS21-28,NJD1-12,ERF3

Amplification results of ERF3 on DRS21-28, NJD1-12 samples.

Click here for additional data file.

Supplemental Information 53 NJD13-32,ERF3

Amplification results of ERF3 on NJD13-32 samples.

Click here for additional data file.

Supplemental Information 54 NJD33,LS1-13,PTU1,LSD1,DGD24-26,ERF3

Amplification results of ERF3 on NJD33, LS1-13, PTU1, LSD1, and DGD24-26 samples.

Click here for additional data file.

Supplemental Information 55 DGD1-23,ERF3

Amplification results of ERF3 on DGD1-23 samples.

Click here for additional data file.

Supplemental Information 56 DGD24-26,ERF3

Amplification results of ERF3 on DGD24-26 samples.

Click here for additional data file.

Supplemental Information 57 LGD1-8,DRS9-20,KNOX-1

KNOX-1 amplification results for LGD1-8, DRS9-20 samples.

Click here for additional data file.

Supplemental Information 58 DRS21-28,NJD1-12,KNOX-1

KNOX-1 amplification results for DRS21-28, NJD1-12 samples.

Click here for additional data file.

Supplemental Information 59 NJD13-32,KNOX-1

KNOX-1 amplification results for NJD13-32 samples.

Click here for additional data file.

Supplemental Information 60 NJD33,LS1-13,PUD1,LSD1,KNOX-1

KNOX-1 amplification results for NJD33, LS1-13, PUD1, LSD1 samples.

Click here for additional data file.

Supplemental Information 61 DGD1-20,KNOX-1

KNOX-1 amplification results for DGD1-20 samples.

Click here for additional data file.

Supplemental Information 62 DGD21-26,KNOX-1

KNOX-1 amplification results for DGD21-26 samples.

Click here for additional data file.

Supplemental Information 63 LGD1-8,DRS9-23,KNOX-2

KNOX-2 amplification results for LGD1-8, DRS9-23 samples.

Click here for additional data file.

Supplemental Information 64 DRS24-28,NJD1-18,KNOX-2

KNOX-2 amplification results for DRS24-28, NJD1-18 samples.

Click here for additional data file.

Supplemental Information 65 NJD19-33,LS1-6,KNOX-2

KNOX-2 amplification results for NJD19-33, LS1-6 samples.

Click here for additional data file.

Supplemental Information 66 LS7-13,PTD1,LSD1,DGD1-13,KNOX-2

KNOX-2 amplification results for LS7-13, PTD1, LSD1and DGD1-13 samples.

Click here for additional data file.

Supplemental Information 67 DGD14-26,KNOX-2

KNOX-2 amplification results for DGD14-26 samples.

Click here for additional data file.

Supplemental Information 68 LGD1-8,DRS9-23,KNOX-3

KNOX-3 amplification results for LGD1-8, DRS9-23 samples.

Click here for additional data file.

Supplemental Information 69 DRS24-28,NJD1-18,KNOX-3

KNOX-3 amplification results for DRS24-28, NJD1-18 samples.

Click here for additional data file.

Supplemental Information 70 NJD19-33,PTD1,LSD1,LS1-4,KNOX-3

KNOX-3 amplification results for NJD19-33, PTD1, LSD1, LS1-4 samples.

Click here for additional data file.

Supplemental Information 71 LS5-13,DGD1-13,KNOX-3

KNOX-3 amplification results for LS5-13, DGD1-13 samples.

Click here for additional data file.

Supplemental Information 72 DGD14-26,KNOX-3

KNOX-3 amplification results for DGD14-26 samples.

Click here for additional data file.

Supplemental Information 73 LGD1-8,DRS9-23,MADS-1

Amplification results of MADS-1 on LGD1-8, DRS9-23 samples.

Click here for additional data file.

Supplemental Information 74 DRS24-28,NJD1-18,MADS-1

Amplification results of MADS-1 on DRS24-28, NJD1-18 samples.

Click here for additional data file.

Supplemental Information 75 NJD19-NJD33,PTD1,LSD1,LS1-4,MADS-1

Amplification results of MADS-1 on NJD19-NJD33, PTD1, LSD1, LS1-4 samples.

Click here for additional data file.

Supplemental Information 76 LS5-13,DGD1-13,MADS-1

Amplification results of MADS-1 on LS5-13, DGD1-13 samples.

Click here for additional data file.

Supplemental Information 77 DGD14-26,MADS-1

Amplification results of MADS-1 on DGD14-26 samples.

Click here for additional data file.

Supplemental Information 78 LGD1-8,DRS9-23,MADS-4

Amplification results of MADS-4 on LGD1-8 and DRS9-23 samples.

Click here for additional data file.

Supplemental Information 79 DRS24-28,NJD1-18,MADS-4

Amplification results of MADS-4 on DRS24-28, NJD1-18 samples.

Click here for additional data file.

Supplemental Information 80 NJD19-33,PUD1,LSD1,LS1-4,MADS-4

Amplification results of MADS-4 on NJD19-33, PUD1, LSD1, LS1-4 samples.

Click here for additional data file.

Supplemental Information 81 LS5-13,DGD1-13,MADS-4

Amplification results of MADS-4 on LS5-13, DGD1-13 samples.

Click here for additional data file.

Supplemental Information 82 DGD14-26,MADS-4

Amplification results of MADS-4 on DGD14-26 samples.

Click here for additional data file.

Supplemental Information 83 LGD1-8,DRS9-23,ABP1-1

ABP1-1 amplification results for LGD1-8, DRS9-23 samples.

Click here for additional data file.

Supplemental Information 84 DRS24-28,NJD1-18,ABP1-1

ABP1-1 amplification results for DRS24-28, NJD1-18 samples.

Click here for additional data file.

Supplemental Information 85 NJD19-33,LS1-4,ABP1-1

ABP1-1 amplification results for NJD19-33, LS1-4 samples.

Click here for additional data file.

Supplemental Information 86 LS5-13,DGD1-13,ABP1-1

ABP1-1 amplification results for LS5-13, DGD1-13 samples.

Click here for additional data file.

Supplemental Information 87 DGD14-26,ABP1-1

ABP1-1 amplification results for DGD14-26 samples.

Click here for additional data file.

Supplemental Information 88 LGD1-8,DRS9-23,ABP1-3

ABP1-3 amplification results for LGD1-8, DRS9-23 samples.

Click here for additional data file.

Supplemental Information 89 DRS24-28,NJD1-18,ABP1-3

ABP1-3 amplification results of DRS24-28, NJD1-18 samples.

Click here for additional data file.

Supplemental Information 90 NJD19-33,PTD1,LSD1,LS1-4,ABP1-3

ABP1-3 amplification results for NJD19-33, PTD1, LSD1, LS1-4 samples.

Click here for additional data file.

Supplemental Information 91 LS5-13,DGD1-13,ABP1-3

ABP1-3 amplification results for LS5-13, DGD1-13 samples.

Click here for additional data file.

Supplemental Information 92 DGD14-26,ABP1-3

ABP1-3 amplification results for DGD14-26 samples.

Click here for additional data file.

Supplemental Information 93 Delat K

When K=3, the value of delat K is the largest and best fit three genetic groups. When K=5, the value of delat K is also large, so we analyze the cases of K=3 and K=5.

Click here for additional data file.

Supplemental Information 94 Gst and Nm between NJD sub-populations

Nm represents the gene flow, and Gst represents the genetic differentiation coefficient.

Click here for additional data file.

Supplemental Information 95 DNA quality test chart of DGD1-6 samples

Two repetitions per sample.

Click here for additional data file.

Supplemental Information 96 DNA quality test chart of DRS3-4, 7-11 samples

Two repetitions per sample.

Click here for additional data file.

Supplemental Information 97 DNA quality test chart of DGD7-14 samples

Two repetitions per sample.

Click here for additional data file.

Supplemental Information 98 DNA quality test chart of DGD18-26 sample

Two repetitions per sample.

Click here for additional data file.

Supplemental Information 99 DNA quality test chart of DRS12-18 sample

Two repetitions per sample.

Click here for additional data file.

Supplemental Information 100 DNA quality test chart of LGD1-8 sample

Two repetitions per sample.

Click here for additional data file.

Supplemental Information 101 DNA quality test chart of DGD15-20 sample

Two repetitions per sample.

Click here for additional data file.

Supplemental Information 102 NJD1, 19, 24, 31-35, LS12 samples DNA quality test chart

Two repetitions per sample.

Click here for additional data file.

Supplemental Information 103 DNA quality test chart of NJD1-8 samples

Two repetitions per sample.

Click here for additional data file.

Supplemental Information 104 DNA quality test chart of LS7-14 samples

Two repetitions per sample.

Click here for additional data file.

Supplemental Information 105 NJD 25-30 samples DNA quality test chart

Two repetitions per sample.

Click here for additional data file.

Supplemental Information 106 NJD 5,11,33,34,35,15 samples DNA quality test chart

Two repetitions per sample.

Click here for additional data file.

Supplemental Information 107 DNA quality test chart of DRS 13-24 samples

Two repetitions per sample.

Click here for additional data file.

Supplemental Information 108 DNA quality test chart of samples LS1-6

Two repetitions per sample.

Click here for additional data file.

Supplemental Information 109 102 sample clustering map raw data

Raw data format when generating a sample cluster map.

Click here for additional data file.

Supplemental Information 110 Population clustering analysis raw data

Generate raw data for population clustering analysis.

Click here for additional data file.

We thank Qing Zhang of Shandong Agricultural University for assistance with the experimental methods.

Additional Information and Declarations

Competing Interests

Author Contributions

Data Availability

The authors declare there are no competing interests.

Yi Wang conceived and designed the experiments, performed the experiments, analyzed the data, prepared figures and/or tables, authored or reviewed drafts of the paper, and approved the final draft.

Yan Ma conceived and designed the experiments, analyzed the data, prepared figures and/or tables, authored or reviewed drafts of the paper, and approved the final draft.

Bingyu Jia performed the experiments, prepared figures and/or tables, and approved the final draft.

Qichao Wu performed the experiments, prepared figures and/or tables, collection of experimental samples, and approved the final draft.

Dekui Zang conceived and designed the experiments, authored or reviewed drafts of the paper, and approved the final draft.

Xiaoyan Yu performed the experiments, authored or reviewed drafts of the paper, and approved the final draft.

The following information was supplied regarding data availability:

Raw data is available in the Supplementary Files.

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
