# Peer review of "Analysis of the genetic diversity of the coastal and island endangered plant species Elaeagnus macrophylla via conserved DNA-derived polymorphism marker"

_PeerJ, doi:10.7717/peerj.8498_

## Round 0.1 · original submission · Major Revisions

Dear authors, we received reports from three reviewers, and all of them agreed that the manuscript is well-written. However, there is a long list of proposed changes needed to be made before the manuscript can be accepted. First of all, you need to implement standard population genetics analysis tools, as recommended by reviewers. I recommend that you pay a special attention to the detailed list of recommendations made by Reviewer 3. Reviewer 2 has provided the annotated manuscript - please use the suggestions to improve your manuscript. I hope that you will be able to implement the required changes in the provided time frame.

Reviewer 1 ·

Basic reporting

The Ms on Analysis of Genetic diversity in the coastal ... By CDDP markers by Wang Yi et al has been reviewed here.
The article is well written. But few technical/methodical lacunae need to be addressed to improve the quality of the study/manuscript.
1. Authors did an island-based population study, but the 'STRUCTURE' analysis and 'Loci selection' is missing. This may be included.
2. There are thousands of Conserved domain genes but authors used only few TF sequences to derive CDDP primers! Authors need to highlight the basis of selecting them.
3. Fig. 2 on DNA extraction, is not very important figure for the Ms. May be deleted.
4. Authors have provided innumerable numbers of supplementary files. May restrict to only few important ones.
Major revision is needed on the above points.

Experimental design

As mentioned earlier: STRUCTURE and Loci selection analysis needed for population study.

Validity of the findings

Mentioned in first section.

Additional comments

Article must be rechecked thoroughly and added more scientific contents to make it a relevant study for the readers.

Reviewer 2 ·

Basic reporting

The key point in this article is that Elaeagnus macrophylla is an endangered species and to protect the biodiversity, its genetic diversity is to be ascertained. However, the author failed to provide any appropriate suggestion, mainly because of the study was ill-conceived and several flaws in analyses of data.

Experimental design

The choice of the marker is not appropriate. Analyses should have also included STRUCTURE analyses.

Validity of the findings

The statements in lines 188-189 and 202-203 are contrary.
Again, the statements that sea current and hydrological profiles alter genetic structure is also too speculative without any valid references which explain that such things do happen

Annotated reviews are not available for download in order to protect the identity of reviewers who chose to remain anonymous.

Reviewer 3 ·

Basic reporting

The manuscript entitled “Analysis of genetic diversity in the coastal and island endangered plant Elaeagnus macrophylla by conserved DNA-derived polymorphism markers (#38915)” presents a molecular marker based study of this species in China, concerning its diversity and population structure.
Authors assessed conserved genes information using CDDP markers and, according to them, low levels of genetic diversity were found for the species. In addition, they reported strong population structure among sampling localities. However, no correlation between genetic distance and geographical distribution were established.
Raw data was supplied and helped to understand authors findigs.
The text presented by authors sounds grammatically ok. However, I’m concerned about the scientific writing and sequence of ideas presented. I’ve found some elements of discussion that may be well suited for a introduction. Also, there are some parts of the results that sounds more familiar for a methodological statement or for the discussion (I’ve pointed these issues below).
Figures must be reviewed. Some of them can be omitted and other must be improved due to the poor quality.

Experimental design

The research fits PeerJ scope and its questions are well defined and meaningful. However, some concerns about methods must be checked by authors. They are listed in “Comments for the author”.

Validity of the findings

This research brought important findings concerning biological dispersion of E. macrophylla and its genetic diversity in the sampled area. Also, the lack of genetic diversity studies of island populations for this species justifies the research. However, there are important caveats that must be addressed before accepting this manuscript. So it is my opinion that the manuscript should not be accepted for publication in PeerJ at this time.

Additional comments

1. Introduction: readers need more information concerning biological issues of E. macrophylla. Ecological and reproductive information of the species are needed to understand the results. Whats the mating system? Whats the main pollinators? How does the dispersion happen? How well this plant is fitted to the coastal/island ambient? Its important to know if a plant uses self or cross-fertilization to understand the diversity data based on molecular makers. Selfing may naturally decreases the magnitude of indexes like heterozygosity in plants and, this decrease may not necessarily represent a poor diversity for some species. So its important to give the elementary biological information of the species for readers understand your results.

2. There are some questions about CDDP markers that must be addressed. As other papers suggested, authors justifies the use of this maker due to its reproducibility when compared with other dominant markers (e.g., RAPD and ISSR). However, even the CDDP developers (Collard and Mackill, 2009) warned that reproducibility problems also exists for CDDP, “suggesting that primer length and annealing temperature per se do not ensure reproducibility”. They recommend gel scoring based on at least two independent replicates of PCRs. It is not clear if authors were careful enough considering the reproducibility problems. Reading the text, there are no elements discussing it. I’m concerned about the confidence of data using markers like this one, mainly when authors states that “weak but distinguishable bands” were scored after electrophoresis. Whats a “weak” band? How to decide when a band is “weak”? Replicates helps to improve the information quality and must be considered during the experimental procedure.

3. As soon as authors described the method for DNA isolation and mention that the DNA is good on quality and in with sufficient amount, I believe that Figure 2 could be omitted. I suggest that authors provide figures concerning DNA isolation as supplemental files.

4. Shannon's information index was presented in Material and Methods section and in Table 3. However, it was not described in the results and neither was discussed. Please, properly present this result and discuss it. Or, remove it from the manuscript.

5. Figure 1 is poor in quality and must be improved. Its resolution is not good enough. We can barely see the province names (e.g., DGD, LS and LSD). We can also barely see the “islands” for the most of sampling localities. I cannot see that Liugong is an island in Figure 1. Are Da Rushan and Laoshan mainland areas? There are too much area in this figure that is not important for the manuscript comprehension. This parts of the figure may be omitted. I don’t think that the whole China map must be present (at least in this size). Emphasizes only what is important for readers understand the main findings.

6. Some parts of the results presentation is only based on showing maximum and minimum values of the estimates. Authors mention measures like averages, but the findings description is focused on the exposition of extreme values. E.g., check lines 174-179. It is important to provide the descriptive statistics of the data and show the general aspects of the genetic diversity. What authors can say about the diversity of populations that didn’t show extreme values?

7. Based on what authors use 0.94 (genetics similarity) to split groups in Figure 4? Despite of the well defined separation of some populations, it is important to mention that the figure scale starts at the 0.88 level. For a more reliable definition of groups it is important to use re-sampling methods in clustering analysis, like bootstrap. Do the authors tried to use this approach? I also suggest using other methods to find out more reliable estimates of number of groups. E. g., Bayesian inference implemented in Structure software (it goes well with dominant data). Or a Discriminant Analysis of Principal Components (implemented in Adegenet package of R). The latter is also preferred to PCoA analysis used by authors due to some particularities of the genetic data. Despite of many references found in literature use PCoA, the percentage of variation explained by the first and second Coordinates is very limited when using molecular markers. It means that with PCoA we loose too much information when we consider only the two dimension plot in analysis. However, if the authors decide to keep using PCoA because they have got high percentage of the variation explained by the two first coordinates, please, show this result.


Specific comments:
1. Line 48: A question: authors evidenced that the species distribution include Jiangsu Province. However this province was not included in the research. Is there an specific reason for not including this province, since it was in the middle of the other sampling localities?
2. Lines 49-51: Please, insert a reference for this statement;
3. Line 52: Please, insert a reference for this statement;
4. Line 68: replace “prime” by “primer”;
5. Line 112: As authors have not used a precise method for DNA quantification, how could they guarantee that they used 30 ng/µL in PCR reactions? If no method like spectrophotometer was used, it is important to mention that this is an approximated measure;
6. Line 128: Does Hs stand for “interspecific” genetic diversity? Isn’t it “intra”? Or better, for authors case, “local genetic diversity”? Please, consider checking it because a single species is being considered in the manuscript.
7. Line 133: R is considered a computational language environment. Is important to mention what kind of analysis was performed using R in methods section. Also, if authors have used specific packages, it is also important to mention them giving their respective references. Include also the R reference (R core team, year).
8. Lines 174-179: estimates of genetic distance and similarity are inversely proportional. Authors don’t have to show both of them. In my opinion it is important to choose one estimate and use it alone so there will be no redundancy.
9. Line 200: Is “PTS” population right? Isn’t it “PTD”?
10. Lines 208-216: Statements presented in these lines look like a discussion. So, authors have to give these information in the proper section.
11. Lines 245-247: Unlike authors affirm, the subpopulation pattern found in Figure 4 disagree with the grouping pattern found in PcoA (Figure 6).
12. Lines 259-264: It is very hardy to compare CDDP marker results with microsatellite (SSR) results when discussing diversity indexes. The letter is codominant and multiallelic, which means that we always expect much more information from SSR than CDDP. A dominant marker like CDDP is interpreted as a bialellic marker and the highest level of heterozygosity expected is 0,5 (or 2pq). For SSR, higher values (e.g., above 0.9) can be found due to the huge amount of alleles found. What I mean is that for a biallelic marker, Hs = 0.5 represents high diversity. However, for SSR, 0.5 of hetherozygosity may represent low level of diversity. Also, comparisons between CDDP and ISSR should be interpreted with caution, since ISSR are potentially not affected by selection. On the other hand, CDDP is developed under the information of functional genes and, naturally, we expect lower levels of variation for these genome regions. Authors suggested in Line 294 that “results of different molecular markers are not fully comparable”, even so they compared results of different markers with no concerning about the main differences among them.
13. Line 309-311: Values of bird propagation distance reported in discussion (480-680 m) are based on unpublished data described in Chung et al., (2002). These distances were poor discussed in Chung et al. The article only cited this values in the INTRODUCTION section and they are not part of a empiric result. Even according to these authors, these values suggests "that seeds are dispersed OVER LONG distances by birds". In addition, this paper brought information about Neolitsea sericea species. I’m not sure if these findings are fully comparable to those found in the present manuscript. I believe that the dispersion by birds should be better explored by authors even with the small gene flow found among localities.
14. Lines 315-317: (a) Angelique (2000) is not listed in References. (b) Whats the specific dispersion information for E. macrophyta? The Angelique (2000) information is for this species? If the ocean is the main responsible for dispersion in E. macrophyta, isn't the species well fitted to this situation? As a reader I need more information to understand this issue because gene flow is a very important issue in your article that must be better explored.

---

## Round 0.2 · Major Revisions

There are several issues, both minor and major. Minor: please format your paper based on PeerJ standards, structure and sections are appropriate. Make sure that the references are in correct format as well. Long sentences need to be spit. Major issues: materials and methods need to contain sufficient information to reproduce the results. Sampling strategy and clustering procedure must be described in details. Selection of the optimal K is a huge area of research, please explain your choice of K.

Please modify your paper and present point-by-point response to the critique by the Reviewer. I hope that after this effort your paper will be acceptable for publication.

Reviewer 3 ·

Basic reporting

The manuscript entitled “Analysis of genetic diversity in the coastal and island endangered plant Elaeagnus macrophylla by conserved DNA-derived polymorphism markers (#38915)” presents a molecular marker-based study of this species in China, concerning its diversity and population structure and is being reconsidered for publication in PeerJ.
There are significant improvements in this version, mainly considering the discussion and the implementation of new statistical analysis. Authors highlighted the main biological concerns related to their findings, which is a very important point to be included in the discussion. However, some questions listed in “General comments” of this review are still present and must be addressed before considering the manuscript publication.

Experimental design

As mentioned in the last review, the research fits PeerJ scope and its questions are well defined.

Validity of the findings

The study found low genetic diversity and high genetic differences among the populations surveyed. Authors claim that the lack of studies for E. macrophylla is the main reason for this study. The research is also justified by its ecological and commercial potential.

Additional comments

a) Authors must check the PeerJ instructions for citing references. There are some references cited (mainly in discussion) that doesn’t fit the instructions. E.g.:
•Line 236: “Weller et al., 1996”; According to the journal rules, this paper has to be cited as “Weller, Sakay & Straub, 1996”. Please, check all the “three-authors” references.
•Lines 314-316: “Leng's research also mentioned that when the seeds of Ilex integra were soaked in seawater, the germination rate was approximately zero (Leng et al., 2005).” This isn’t the most appropriate way to cite a reference. I suggest something like “Leng et al. (1995) also mentioned that when the seeds of Ilex integra were soaked in seawater, the germination rate was approximately zero.” … or … “Seeds of Ilex integra also presented germination rate close to zero when soaked in seawater (Leng et al., 2005).” There are several references cited in this way within the Discussion.

b) Lines 77-82: This statement is too long. Breaking the sentence may provide a clearer information. Also, correct the “of and” at the end of the line 77.

c) Lines 132-134: Structure Harvester implements the method described by Evanno et al. (2005; doi: 10.1111/j.1365-294X.2005.02553.x). Authors have to insert this reference in the text since they have used the method.

d) Lines 162-163: This is a “discussion” statement and should not be descried within “results” section.

e) Lines 180-195: It's still not clear which criteria the authors used to split the samples in the different groups. I can see that the “considered groups” are based in the main branches observed in the dendrogram. However, this is a little arbitrary. Are these "groups" consistent? Based on what the authors know that they really have “true groups” in this analysis? Why not presenting the bootstrap to improve the statement? If they used the "cut-line" (e.g., based on the average similarity), it is not showed in Figure 4, neither in the “Material and Methods” section. It's important to make it clear, so the reader will know the basis of the method used to split the samples in different groups. Another way to avoid this problem is bringing the focus of the statement for the fact that no cross-clustering were observed among samples of different locations. Regardless of this information is present in lines 189-190, its sounds like a “secondary” information in the text.

f) Lines 199-214: Considering that authors have used the Evanno et al. (2005) criteria (delta K) to detect the most probable number of clusters, the results of delta K must be showed in the manuscript. Authors showed values for K=3 and K=5. But I cannot evaluate if these solutions are the best ones for the samples since I don’t have the magnitude of delta-K values. Please, provide delta-K values or the graph showing the likelyhood for each tested K value. E. g., authors said that "when K = 5, the value of delta K is also large”. But, how large is it?

g) Line 248: “delicious” is a very relative terminology. I’m not sure if Zang (2016) have really used this word but, in general, it is not well suited in scientific language.

h) Line 330: It’s interesting that Gst values indicates that NJD can be arranged in two subpopulations. However, this information was not presented in the Results section. It is also important to mention this approach in Methods, saying that the Gst index were calculated within localities when significant differences among specimens were found.

i) Include the DOI in the references listed.

---

## Round 0.3 · Minor Revisions

Thank you very much for making the requested changes. At this time, there are several minor issues have been identified. Fixing this issues will improve the presentation of material. In addition, the reviewer expressed concerns over the interpretation of the population structure analysis. May be, it will benefit your paper if you add the discussion of the existence of three main genetic groups. Please refer to the detailed comments by the reviewer.

Reviewer 3 ·

Basic reporting

The manuscript entitled “Analysis of genetic diversity in the coastal and island endangered plant Elaeagnus macrophylla by conserved DNA-derived polymorphism markers (#38915)” presents a molecular marker-based study of this species in China, concerning its diversity and population structure.
In this third review round we still have some minor questions to be addressed (see “General comments”), but, at this time, the manuscript presents most of the necessary elements to be accepted.

Experimental design

As mentioned in the first review, the research fits PeerJ scope and its questions are well defined.

Validity of the findings

The study found low genetic diversity and high genetic differences among the populations surveyed. Authors claim that the lack of studies for E. macrophylla is the main reason for this study. The research is also justified by its ecological and commercial potential.

Additional comments

Lines 124-125
Replace:
… and gene flow (Nm=1-Gst/Gst) among populations or localities when significant differences among specimens were found.
By:
… and gene flow (Nm=1-Gst/Gst) among populations. The estimates were also calculated within sampling localities when significant differences among specimens were found.


Lines 134-137
Please, revise this statement considering its grammatical improvement
Compress the result of the operation (".zip" format) and upload it to the Structure Harvester website (http://ztaylor0.bi-ology.ucla.edu/structure) to get the analysis result of the K value (Evanno, Regnaut & Goudet, 2005.)


Lines 227-230
Replace:
Frankham compared and analyzed the allelic diversity of 202 groups of land and island
populations of various species, including mammals, birds, fish, reptiles, insects and plants; in
165 cases (81.7%), the genetic diversity of island populations was lower than that of terrestrial
populations, with an average decrease of 29% (Frankham, 1997)
By:
Frankham (1997) compared and analyzed the allelic diversity of 202 groups of land and island
populations of various species, including mammals, birds, fish, reptiles, insects and plants; in
165 cases (81.7%), the genetic diversity of island populations was lower than that of terrestrial
populations, with an average decrease of 29%.


Line 288
… clustering analysis shows that there was no hybridization among individuals
Isn’t it supposed to be “there was no hybridization among individuals from different localities”?


About Delta-K values
Despite authors have answered in rebuttal letter that the Delta-K values were provided in supplemental files, I believe it could be presented within Figure 5 (e.g., Fig. 5B). This result is important to prove how good are the model chosen by authors after Bayesian analysis.
Besides, I also have tow additional comments/questions about Structure results:
a) It’s clear that Structure pointed that E. macrophylla genetic diversity can be arranged into three main genetic groups. There was also a substructure found considering 5 genetic groups. Authors described very well the importance of the 5 subgroups. However, in discussion, the 3 main genetic groups were neglected. What are the main conclusions about these 3 main genetic groups?
b) The occurrence of subgroups in NJD population was well described by authors. They also repeated some analysis considering only this samples (line 216; Gst and Nm of the two sub-populations of NJD). However, its clear for me that they also have another subpopulation structure in DRS samples. It can be observed in Figure 5 and, also, in the dendrogram. Part of this specific result for DRS is mentioned briefly in results, while in NJD we have an specific section to show this findings (line 216; Gst and Nm of the two sub-populations of NJD). A discussion about this result is also absent in the manuscript.

---

## Round 0.4 · Minor Revisions

I have no more questions about the scientific content of the manuscript. Some parts of the paper may benefit from additional proofreading by a native English speaker. I have annotated the manuscript with my suggestions.

---

## Round 0.5 · Minor Revisions

Dear authors, the paper looks very much improved. I have just a couple of final suggestions. First, I recommend to check declaration of every abbreviation in the text. It is better not to introduce and not to use abbreviations in the abstract. You need to define each acronym only once (some of them are defined several times in your texts). Second suggestion is about the quality of written English. Overall, the manuscript is well written. However, the placement of "a" and "the" articles appears to be almost random. Please show your manuscript to a native speaker.

---

## Round 0.6 · Minor Revisions

I have to hold your manuscript in the minor revision stage. I saw that you have implemented some of the reviewer's suggestions, but I could not find where you have addressed these comments:

"a) It’s clear that Structure pointed that E. macrophylla genetic diversity can be arranged into three main genetic groups. There was also a substructure found considering 5 genetic groups. Authors described very well the importance of the 5 subgroups. However, in discussion, the 3 main genetic groups were neglected. What are the main conclusions about these 3 main genetic groups?
b) The occurrence of subgroups in NJD population was well described by authors. They also repeated some analysis considering only this samples (line 216; Gst and Nm of the two sub-populations of NJD). However, its clear for me that they also have another subpopulation structure in DRS samples. It can be observed in Figure 5 and, also, in the dendrogram. Part of this specific result for DRS is mentioned briefly in results, while in NJD we have an specific section to show this findings (line 216; Gst and Nm of the two sub-populations of NJD). A discussion about this result is also absent in the manuscript."

I have also attached a document with suggested corrections. Please show your manuscript to a native English speaker or a technical editor.

---

## Round 0.7 · accepted · Accept

Thank you very much for making the requested suggestions. The manuscript has been significantly improved.